# Spiking PointNet: Spiking Neural Networks for Point Clouds

**Dayong Ren, Zhe Ma, Yuanpei Chen, Weihang Peng, Xiaode Liu, Yuhan Zhang, Yufei Guo**[*]
Intelligent Science & Technology Academy of CASIC, China
Scientific Research Laboratory of Aerospace Intelligent Systems and Technology, China
`rdyedu@gmail.com, yfguo_bit@126.com, yfguo@pku.edu.cn`

## Abstract

Recently, Spiking Neural Networks (SNNs), enjoying extreme energy efficiency, have drawn much research attention on 2D visual recognition and shown gradually increasing application potential. However, it still remains underexplored whether SNNs can be generalized to 3D recognition. To this end, we present Spiking PointNet in the paper, the first spiking neural model for efficient deep learning on point clouds. We discover that the two huge obstacles limiting the application of SNNs in point clouds are: the intrinsic optimization obstacle of SNNs that impedes the training of a big spiking model with large time steps, and the expensive memory and computation cost of PointNet that makes training a big spiking point model unrealistic. To solve the problems simultaneously, we present a trained-less but learning-more paradigm for Spiking PointNet with theoretical justifications and in-depth experimental analysis. In specific, our Spiking PointNet is trained with only a single time step but can obtain better performance with multiple time steps inference, compared to the one trained directly with multiple time steps. We conduct various experiments on ModelNet10, ModelNet40 to demonstrate the effectiveness of Spiking PointNet. Notably, our Spiking PointNet even can outperform its ANN counterpart, which is rare in the SNN field thus providing a potential research direction for the following work. Moreover, Spiking PointNet shows impressive speedup and storage saving in the training phase. Our code is open-sourced at Spiking-PointNet.

## 1 Introduction

The advent of deep learning technologies, notably PointNet [38], has considerably amplified our capabilities to comprehend and manipulate intricate 3D data from real-world settings. With autonomous driving and augmented reality, which often require real-time interaction and fast response, becoming increasingly prevalent, the reliance on efficient point cloud processing techniques has been escalated. However, computation for the point cloud is energy-hungry and usually needs powerful devices.

Spiking Neural Networks (SNNs) [40; 4; 11; 12; 39; 35; 2; 55; 22; 57; 56; 47; 52; 44; 53; 54], seen as more energy efficient than Artificial Neural Networks (ANNs) due to their event-driven computation mechanism and the energy-saving multiplication-addition transformation advantage, have received extensive attention recently in many fields. For example, in [36], SNNs were used to handle sequential learning and show better performance and less energy cost on sequential learning compared to ANNs with similar scales. In [31], SNNs were leveraged to study the Human Activity Recognition (HAR) task. The results show that the SNN can reduce up to 94% energy consumption while being comparable to homogeneous ANN counterparts in accuracy. There are also some works that apply SNNs in autonomous driving. LaneSNNs [45] presented an SNN-based approach to detect

---

[*]Corresponding author.

37th Conference on Neural Information Processing Systems (NeurIPS 2023).

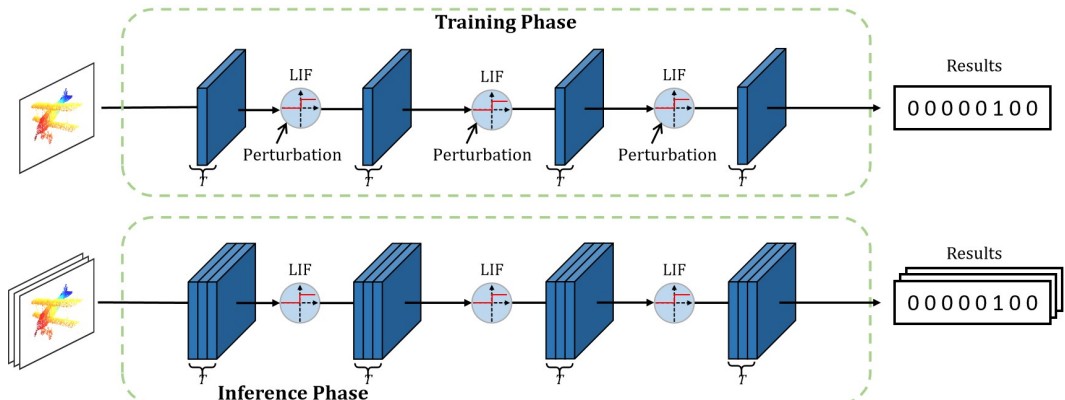

Figure 1: The overall of the trained-less but learning-more framework. The Spiking PointNet is trained with only one single time step in the training phase, while is used with multiple time steps in the inference phase. To improve the performance of the SNN, we also add some membrane potential perturbation in the training.

the lanes with an event-based camera input with a very low power consumption of about 1 W. For the more challenging point cloud task, a question is naturally raised: Could SNNs be transferred to the 3D domain and retain the energy-efficient advantage?

To this end, we present **Spiking PointNet**, the first spiking neural network approach to deep learning on point clouds. To better apply the SNNs in the point cloud field, we focus on solving two huge obstacles staying on this road. The first is optimizing difficulty. Though the binary spike information transmission paradigm makes SNNs much energy efficient, it also introduces the training challenge since the gradients for firing procession of the spiking neuron are not well-defined, and they are all almost zero or infinite sometimes. The zero-but-all gradient makes it impossible to train SNNs via gradient-based optimization methods like ANNs. To handle this problem, various Surrogate Gradient (SG) methods have been proposed [35; 49; 39; 30; 14]. This kind of method tries to find an alternative function to replace the firing function when doing back-propagation of the spiking neurons. Thus, the SNN can be also trained with the current gradient-based optimization framework. However, it is not easy to find a suitable surrogate function, especially for these SNNs with large time steps. With the increasing of time steps, the explode or vanish problem and the gradient error problem will be severe. We will provide a detailed analysis in Sec. 3.3.

The second problem is that training networks for point clouds need more expensive memory and computation than images since point cloud data requires more dimensions to describe itself. To overcome this limitation in point clouds, researchers have proposed various model simplification strategies. These strategies include but are not limited to, sparse convolution [7], optimization during the data processing phase [27], and optimization at the local feature extraction stage [34; 33]. However, for applying the SNN to point clouds, the memory and computation will be enlarged greatly still with the increasing of time steps, and the above methods cannot handle this problem well. Thus, there is no existing way to train SNNs with large time steps on common deep-learning devices.

To solve the above problems simultaneously, we present a trained-less but learning-more paradigm for Spiking PointNet. Specifically, we propose a new framework for Spiking PointNet, that we train the SNN using a suitable SG method with only a single time step and infer it with multiple time steps to obtain a better performance. We will prove theoretically and experimentally that this framework can result in a better SNN than training it with multiple time steps directly in Sec. 3.4. To improve the framework further, we also embed a membrane potential perturbation method in the framework based on the observation that the residual membrane potential of SNN coming from the previous time step cannot transmit the temporal information for static point cloud datasets but a perturbation to increase the generalization. The overall workflow of the framework is visualized in Fig. 1.

The contributions of our paper are as follows:

- We prove that it is not easy to train a well-performed SNN with large time steps directly for point clouds with theoretical justifications and in-depth experimental analysis and propose

the Spiking PointNet with a trained-less but learning-more framework, a first simple yet effective SNN framework for point clouds.

- Furthermore, we also propose a membrane potential perturbation method for the framework to increase the SNN generalization.

- We evaluate our methods on various datasets and the experimental results show the effectiveness of our method. Rather, our Spiking PointNet even can outperform its ANN counterpart, which is very rare in the SNN field.

## 2 Related Work

### 2.1 Spiking Neural Networks

Generally, there are three kinds of methods to train SNNs [16]: (1) spike-timing-dependent plasticity (STDP) [1] approaches, (2) ANN to SNN conversion approaches [25; 24; 32; 8; 10; 3; 22; 29], and (3) directly training approaches [6; 35; 49; 39; 30; 46; 47; 18; 21; 17; 13]. STDP is a kind of biology-inspired method [23; 9] that updates the weights with the unsupervised learning algorithm called Hebbian learning [43]. However, it is limited to small-scale datasets yet. The ANN-to-SNN conversion [8; 29] converts a well-trained ANN checkpoint to the SNN counterpart. Since training an ANN is much faster than training an SNN, this kind of method provides a fast way to obtain an SNN without using gradient descent for SNNs at all. However, it does not have its own learned feature. In specific, all the converted SNN does is to mimic the ANN. Moreover, this type of method requires many time steps to obtain a high-accuracy SNN. The direct training method tries to find an alternative function to replace the firing function of the spiking neurons when doing back-propagation. This kind of method can narrow the time steps greatly, even less than 5 [20; 15; 14], hence has received much attention recently. However, it is not easy to find a suitable surrogate function for these SNNs with large time steps. In this work, we focus on solving the problem.

### 2.2 Deep Learning on Point Clouds

Training networks for point clouds need expensive memory and computation. To address the challenges posed by expensive computation and memory requirements, researchers have proposed a series of model simplification strategies to overcome the limitations of current point cloud models in practical applications [7; 41; 27; 34; 33; 42]. For instance, Lee *et al*. [28] introduced PillarAcc, an innovative algorithm-hardware co-design that significantly enhances the performance and energy efficiency of 3D object detection. However, its reliance on complex sparse convolution and dynamic pillar pruning may introduce additional complexity in the design and implementation process. Choy *et al*. [7] proposed MinkowskiConv, which provides a comprehensive solution for handling sparse spatio-temporal data, greatly enhancing its ability to capture complex temporal patterns in the data. Nevertheless, the inherent computational complexity and memory demands of 4D convolutions present new challenges. Hu *et al*. [27] introduced RandLA-Net to conserve computational resources in point cloud analysis by leveraging random sampling and an efficient local feature aggregation module. However, a limitation of RandLA-Net is that random sampling may lead to the loss of critical information and cannot be seamlessly applied to existing networks without a decline in performance. In comparison, the SNN version of PointNet offers an effective solution by significantly improving algorithm execution efficiency without altering the overall network structure, reducing dependence on high-performance devices in the inference. This enables general-purpose networks to more effectively address the computational resource consumption challenges of practical point cloud networks without the need to redesign network structures. However, for applying the SNN to point clouds, the memory and computation will be enlarged greatly still with the increasing of time steps in the training time. And, there is no existing way to train SNNs with large time steps on common deep-learning devices.

## 3 Preliminary and Methodology

In the paper, we mainly apply the SNN for the PointNet [38], the first deep learning model that processes raw point clouds directly, and modify it to the Spiking PointNet. Here, we first introduce the PointNet and widely used SNN neuron model, Leaky Integrate-and-Fire (LIF) model in detail. Then we will elucidate the difficulty of optimizing the Spiking PointNet with large time steps. Next,

a trained-less but learning-more framework to solve the above problem will be presented. Finally, we further improve it with a membrane potential perturbation method.

## 3.1 PointNet

PointNet represents a novel application of deep learning to process point cloud data [38]. It effectively addresses two primary challenges: permutation invariance, the unordered nature of point cloud data, and rotational invariance, the freedom to rotate the point cloud in 3D space without altering the represented object. Specifically, to tackle these challenges, PointNet employs a symmetric function in conjunction with a spatial transformer network. It processes each point through a shared fully connected network, followed by a max pooling operation. This approach inherently ensures permutation invariance as it remains indifferent to the order of input points. Formally, given point cloud data $\{x_1, x_2, ..., x_n\}$, each point $x_i$ is transformed via a shared Multi-Layer Perceptron (MLP) denoted by $h$, followed by a max pooling operation to enforce symmetry, yielding a global feature descriptor. Therefore, PointNet approximates a general function $f$ defined on a point set by applying a symmetric function $g$ on transformed elements in the set:

$$f\left(\{x_1, \ldots, x_n\}\right) \approx g\left(h\left(x_1\right), \ldots, h\left(x_n\right)\right), \tag{1}$$

where $f : 2^{\mathbb{R}^N} \to \mathbb{R}, h : \mathbb{R}^N \to \mathbb{R}^K$ and $g : \underbrace{\mathbb{R}^K \times \cdots \times \mathbb{R}^K}_{n} \to \mathbb{R}$ is a symmetric function.

For rotational invariance, PointNet introduces a spatial transformer network - a specialized neural network proficient at predicting the required spatial transformation matrix for the point cloud, thereby enabling PointNet to manage rotating point cloud data.

The principal divergence between PointNet and conventional point cloud processing methodologies resides in the implementation of deep neural networks. This represents a significant leap from the traditional approach of manually designed features to Artificial Neural Networks (ANNs). The proposed model, Spiking PointNet, advances this progression by transitioning from ANNs to Spiking Neural Networks (SNNs). SNNs, which emulate the neural mechanisms of the brain more closely, promise to enhance the efficiency and precision of point cloud processing outcomes.

## 3.2 Explicitly Iterative LIF Model

SNNs use the spiking neuron, which is inspired by the brain's natural mechanisms, to transmit information. A spiking neuron will receive input spike trains from the previous layer neuron models along times to update its membrane potential, $u$. In the paper, we adopt the widely used leaky integrate and fire (LIF) neuron model, which can be described as follows:

$$\tau_\mathrm{m} \frac{du}{dt} = -\left(u - u_\mathrm{rest}\right) + R \cdot I(t), \quad u < V_\mathrm{th}. \tag{2}$$

In the above equation, $I$ represents the input current, $V_\mathrm{th}$ is the threshold, and $R$ and $\tau_\mathrm{m}$ are the resistance and time constant, respectively. A spike will be generated when $u$ reaches $V_{th}$, and $u$ is subsequently reset to the resting potential $u = u_\mathrm{rest}$, typically set to zero [30; 11; 39].

To use the mature machine learning framework (*e.g.*, TensorFlow, Pytorch) to train the SNNs, an explicitly iterative LIF spiking model was proposed in [49] given by

$$u_i[t + 1] = \lambda\left(u_i[t] - V_\mathrm{th} s_i[t]\right) + \sum_j w_{ij} s_j[t] + b_i,$$

$$s_i[t + 1] = H\left(u_i[t + 1] - V_\mathrm{th}\right). \tag{3}$$

Here, $I_i(t) = \sum_j w_{ij} s_j(t) + b_i$, where the subscript $i$ denotes the $i$-th current neuron, $w_{ij}$ is the weight from $j$-th neuron in the previous layer connected to the current neuron $i$, and $b_i$ is a bias. $H(x)$ signifies the Heaviside step function, $s_i[t]$ is the spike train of neuron $i$ at discrete time step $t$, and $\lambda < 1$ is a leaky term for $1 - \frac{1}{\tau_\mathrm{m}}$, typically is 0.20 or 0.25 as in [30; 6; 39; 20].

The main difference between ANNs and SNNs is the nonlinear computational neuron. Replacing the ReLU neuron from PointNet with LIF spiking neuron will transform the PointNet to Spiking PointNet.

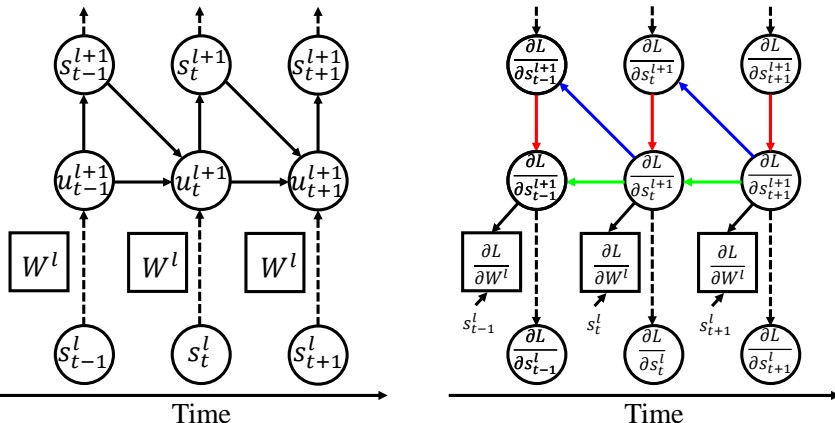

Figure 2: Chain rule graph for gradients w.r.t. weights of SNNs

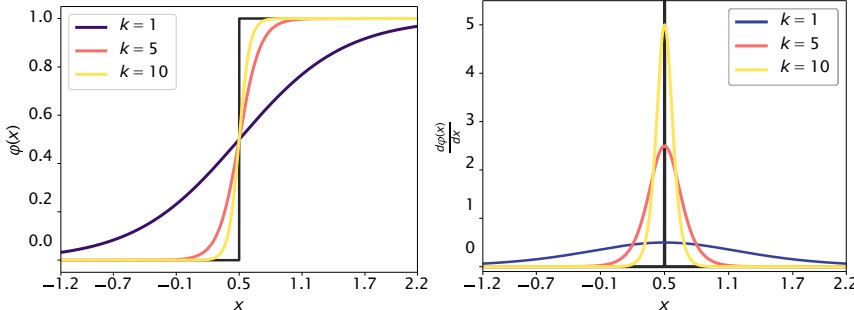

Figure 3: The surrogate function (left) under different values of the coefficient, $k$ and its corresponding gradient (right). The blue curves represent the firing function (left) and its true gradient (right).

### 3.3 Optimizing Difficulty for SNNs with Large Time Steps

A notorious problem in SNN training is the non-differentiability of the firing function, see Eq. (3). To discuss this problem concretely, we denote the loss function as $L$ and calculate the gradients w.r.t. weights using the chain rule following [51] shown in Fig. 2 and given by

$$\frac{\partial L}{\partial \mathbf{W}^l} = \sum_{t=1}^{T} \frac{\partial L}{\partial \mathbf{s}^{l+1}[t]} \frac{\partial \mathbf{s}^{l+1}[t]}{\partial \mathbf{u}^{l+1}[t]} \left( \frac{\partial \mathbf{u}^{l+1}[t]}{\partial \mathbf{W}^l} + \sum_{\tau < t} \prod_{i=t-1}^{\tau} \left( \frac{\partial \mathbf{u}^{l+1}[i+1]}{\partial \mathbf{u}^{l+1}[i]} + \frac{\partial \mathbf{u}^{l+1}[i+1]}{\partial \mathbf{s}^{l+1}[i]} \frac{\partial \mathbf{s}^{l+1}[i]}{\partial \mathbf{u}^{l+1}[i]} \right) \frac{\partial \mathbf{u}^{l+1}[\tau]}{\partial \mathbf{W}^l} \right),$$

(4)

where $\mathbf{W}^l$ represents the weights from layer $l$ to $l+1$, $T$ is the total time steps, and $L$ is the loss. The terms $\frac{\partial \mathbf{s}^l[t]}{\partial \mathbf{u}^l[t]}$ for firing function is non-differentiable. Its gradient is 0 almost everywhere except for the threshold. Therefore, the actual updates for weights would either be 0 or infinity when recalling the gradient descent. To handle this problem, many surrogate gradient methods are proposed [49; 58; 19]. In this kind of method, when performing the forward pass, the firing function remains exactly the same, while, when for the backward pass, the firing function will become a surrogate function, and the surrogate gradient is computed based on it. A typically surrogate function may refer to the `tanh`-like function [14; 5; 30], given by

$$\varphi(x) = \frac{1}{2} \tanh\left(k\left(x - V_{\text{th}}\right)\right) + \frac{1}{2},$$

(5)

where $k$ is a constant. The $\varphi(x)$ and its gradient can be seen in Fig. 3. The surrogate gradient can be adjusted by changing $k$. Other widely used surrogate functions also enjoy the same characteristic, such as rectangular or sigmoid surrogate functions proposed in [49].

It can be seen that, when $k$ is set as a large value, a more accurate gradient in the backward pass can be obtained, *i.e.*, the gradient will be sharp at a narrow range while gradual in the residual part. However, the gradient explode or vanish problem will become more severe in this case since the final

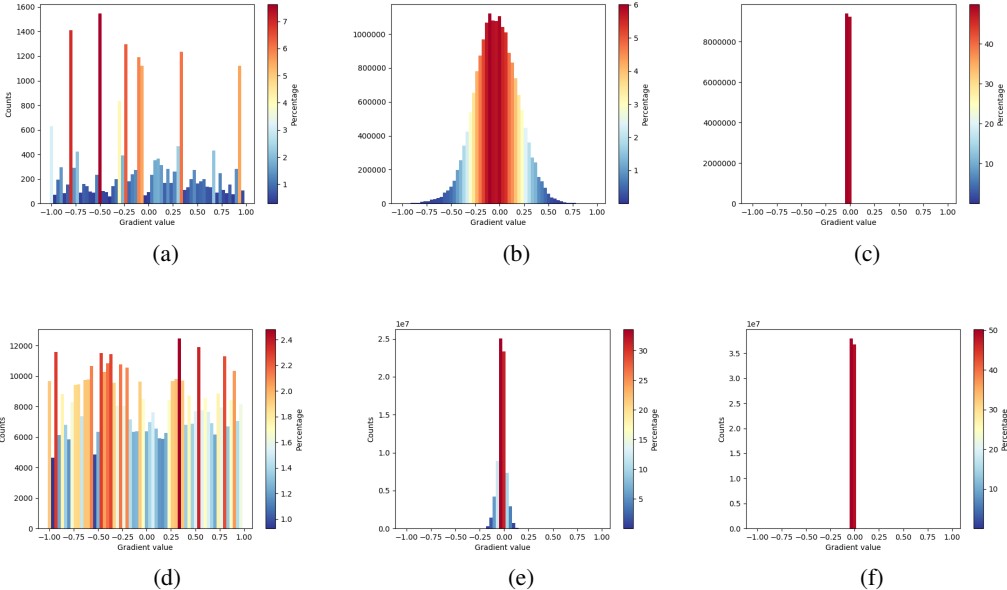

Figure 4: The gradient distributions of the first layer for Spiking PointNet on ModelNet40 with different $k$ and time steps. (a), (b), and (c) show the distributions for the Spiking PointNet using 1 single time steps with $k = 0.5, 5, 20$, respectively. (e), (d), and (f) show the distributions for the Spiking PointNet using 4 time steps with $k = 0.5, 5, 20$, respectively.

weight gradient is calculated by multiplying many surrogate gradients through layers and time steps according to Eq. (4), which tends to be either very big or small. While, when $k$ is set as a small value, a more inaccurate gradient in the backward pass will be obtained [14]. Hence the gradient error will be accumulated through layers and time steps, thus hurting the performance of the SNN too [48]. Consequently, it is very difficult to train a well-performed SNN with large time steps directly, limited by the fact that there is no suitable surrogate gradient for this kind of SNN.

### 3.4 The Trained-less But Learning-more Framework

As aforementioned, except for the optimizing difficulty, there is no existing suitable way to train SNNs with large time steps on common deep-learning devices for point clouds, since training network on point clouds is much energy and memory hungry. To handle these two problems simultaneously, we propose a trained-less but learning-more framework.

To better describe the paradigm, we first show the gradient distributions of the first layer for Spiking PointNet on the ModelNet40 in the Fig. 4. Here, we have several baselines: (1) The Spiking PointNet using 1 single time step along with $k = 0.5, 5, 20$, respectively; (2) the Spiking PointNet using 4 time steps along with $k = 0.5, 5, 20$, respectively. It can be seen that, when $k = 5$, the gradient distribution for Spiking PointNet with 1 single time step is relatively suitable. While $k = 20$, the explode or vanish problem is very significant, and when $k = 0.5$, the distribution is relatively flat, which means it is different from the actual gradient greatly and the gradient error is huge. Hence, a small $k$ or a large $k$ is not a good idea for SNNs. The results in Tab. 1 also show that a small $k$ or a large $k$ will reduce the SNN accuracy.

Nevertheless, we can still find a relatively suitable surrogate function for the SNN with few time steps. However, the explode or vanish problem and the gradient error problem will be more severe with the time step increasing for SNNs. It can be seen that, although the $k = 5$ is a good choice for the Spiking PointNet with 1 single time step, the explode or vanish problem will become very severe for the Spiking PointNet with 4 time steps. Meanwhile, with the time step increasing, the gradient error problem becomes severe too. Note that, when $k = 0.5$, the gradient distribution for Spiking PointNet with 4 time steps becomes flatter, which means a huger gradient error.

Table 1: The accuracy for Spiking PointNet with different time steps and $k$ on ModelNet40.

| Time step | $k$ | | |
|:---:|:---:|:---:|:---:|
| | 0.5 | 5 | 20 |
| 1 | 80.34% | 86.98% | 83.46% |
| 4 | 76.73% | 86.70% | 75.36% |

Consequently, it is not easy to train a Spiking PointNet with large time steps. The Tab. 1 also shows that the Spiking PointNet with 4 time steps even performs worse than the one with only one single time steps. To this end, we propose a trained-less but learning-more framework. In specific, we train our Spiking PointNet with only a single time step but use it with multiple time steps in the inference time. By training SNNs with only one single time step, the gradient explode or vanish problem will be mitigated greatly. Thus we can choose a relatively large $k$, and meanwhile, the gradient error will be reduced at the same time. In the paper, we choose $k$ as 5. The Tab. 2 shows the results of our trained-less but learning-more framework for Spiking PointNet on ModelNet10 and ModelNet40. It can be seen that training the Spiking PointNet with a suitable surrogate function will outperform the one with 4 time steps, and if we infer the trained model with multiple time steps, the accuracy will increase some still. Thus we name the paradigm as the trained-less but learning-more framework.

Table 2: The ablation study for the trained-less but learning-more framework.

| Dataset | Training: 4 T | Training: 1 T | | | |
|:---:|:---:|:---:|:---:|:---:|:---:|
| | Inferring: 4 T | Inferring: 1 T | Inferring: 2 T | Inferring: 3 T | Inferring: 4 T |
| ModelNet10 | 91.05% | 91.99% | 92.43% | 92.53% | 92.32% |
| ModelNet40 | 86.70% | 86.98% | 87.26% | 87.21% | 87.13% |

Training: $n$ T denotes training the Spiking PointNet with $n$ time steps. Inferring: $n$ T denotes Inferring the Spiking PointNet with $n$ time steps.

## 3.5 Membrane Potential Perturbation Method

An interesting phenomenon in our trained-less but learning-more framework is that though the Spiking PointNet is trained with only 1 single time step, in the inference, with the increasing of time steps, the accuracy will increase less or more at the same time. Some work [37; 31] proves that the SNNs can extract spatio-temporal features for sequential data with multiple time steps. However, the point cloud is the static data, thus there is no temporal feature to extract. We guess that the reason for the accuracy increase of Spiking PointNet with multiple time steps is that it becomes an ensemble. The residual membrane potential along time steps in the spiking neuron can be seen as the perturbation. The perturbation will provide different initializations for the Spiking PointNet along time steps. Thus the Spiking PointNet at every time step can be seen as a different model. And averaging their outputs can improve the uncertainty estimation and thus may lead to an enhancement in SNN accuracy.

To verify our guess, in this section, we conducted a series of ablation experiments on ModelNet40. We trained the Spiking PointNet with 4 time steps and evaluated its accuracy at every time step and all time steps respectively. The results are shown in Tab. 3. It can be seen that, the collective results outperform those obtained from individual steps, implying that the performance improvement associated with larger time steps might be more related to an ensemble learning effect, rather than a direct result of the increased time steps. In specific, the Spiking PointNet at each time step can be seen as an independent model casting a vote towards the final prediction. This ensemble learning strategy increases the robustness of the model and subsequently improves the prediction accuracy. Our study suggests that a rethinking and optimization of time steps in SNNs is warranted. The inherent ensemble learning effect, which is underappreciated in the conventional SNN design, could be a viable strategy to enhance the performance of SNNs, while also managing computational resources. Our insights provide valuable implications for future design and optimization strategies in the field of SNNs.

Table 3: The verification test for the effect of the time step on the static dataset.

|  | 1-th time step | 2-th time step | 3-th time step | 4-th time step | Averaging all |
|---|---|---|---|---|---|
| Accuracy | 83.70% | 84.65% | 85.70% | 85.29% | 86.70% |

Under the perspective that the residual membrane potential of SNN, coming from the previous time step cannot transmit the temporal information for static point cloud datasets but a perturbation to increase the generalization, we further propose a membrane potential perturbation method for the framework. In specific, we add some membrane potential perturbation randomly to initial the spiking neurons of the Spiking PointNet at each epoch in the training phase, thus the generalization of the model trained with only 1 single time step will be improved like those trained with multiple time steps. The results for the trained-less-based Spiking PointNet with membrane potential perturbation are shown in Tab. 4. It can be seen that with the perturbation method, the Spiking PointNet further gets another performance lift, amounting to 93.31% and 88.61% final accuracy for ModelNet10 and ModelNet40 respectively.

Table 4: The ablation study for the membrane potential perturbation.

| Dataset | Method | Training: 1 T | | | |
|---|---|---|---|---|---|
|  |  | Inferring: 1 T | Inferring: 2 T | Inferring: 3 T | Inferring: 4 T |
| ModelNet10 | without MPP | 91.99% | 92.43% | 92.53% | 92.32% |
|  | with MPP | 91.66% | 92.98% | 92.98% | 93.31% |
| ModelNet40 | without MPP | 86.98% | 87.26% | 87.21% | 87.13% |
|  | with MPP | 87.72% | 88.46% | 88.25% | 88.61% |

MPP denotes membrane potential perturbation.

## 4 Experiments

In this section, we conduct extensive experiments on ModelNet10 and ModelNet40 [50] to demonstrate the superior performance of our method. ModelNet10 and ModelNet40 are two widely recognized public datasets used for 3D object classification, curated and maintained by a research team at Princeton University. ModelNet10 is a compact dataset comprising 4,899 3D models that span 10 distinct categories such as tables, chairs, bathtubs, and guitars. This dataset is a subset of ModelNet40, offering fewer categories but with more pronounced differences between each category. This characteristic makes ModelNet10 an excellent starting point for evaluating the performance of 3D classification algorithms. ModelNet40 is a more comprehensive dataset, containing approximately 12,311 3D models across 40 different categories, including tables, chairs, airplanes, guitars, and more. With an expanded array of categories and samples, ModelNet40 serves as a robust benchmark for gauging the performance of 3D classification algorithms in more complex and challenging tasks. We leverage the PointNet architecture for point cloud classification tasks. For all our SNN models, we set $V_{\text{th}}$ as 0.5, The initial perturbations, $\delta$, range from 0 to 0.5.

### 4.1 Ablation Studies

We first conducted thorough ablation experiments of our method against the vanilla SNN for PointNet on the ModelNet10/40 datasets. The Tab. 5 displays the performances of various methods under different training and testing time steps. On the ModelNet10 dataset, our Spiking PointNet with membrane potential perturbation (MPP) reaches an accuracy of 93.31% with a testing time step of 4, which outperforms both the one without MPP (92.32%) and the ANN-based approach (92.98%). Even with a testing time step of 1, our Spiking PointNet with MPP still achieves an accuracy of 91.66%, surpassing the performance of vanilla Spiking PointNet trained with 4 time steps (89.62%). This validates the effectiveness of our method. Further, on the ModelNet40 dataset, our Spiking PointNet with MPP attains an accuracy of 88.61% with a testing time step of 4, also outperforming

Table 5: Comparison between our method and the vanilla SNN on ModelNet10/40 datasets.

| Datasets | Methods | Training time steps | Testing time steps | | | |
|---|---|---|---|---|---|---|
| | | | 1 | 2 | 3 | 4 |
| ModelNet10 | ANN | - | 92.98% | | | |
| | Vanilla SNN | 4 | 89.62% | 90.83% | 91.05% | 91.05% |
| | Ours without MPP | 1 | 91.99% | 92.43% | 92.53% | 92.32% |
| | Ours with MPP | 1 | 91.66% | 92.98% | 92.98% | 93.31% |
| ModelNet40 | ANN | - | 89.20% | | | |
| | Vanilla SNN | 4 | 85.59% | 86.58% | 86.34% | 86.70% |
| | Ours without MPP | 1 | 86.98% | 87.26% | 87.21% | 87.13% |
| | Ours with MPP | 1 | 87.72% | 88.46% | 88.25% | 88.61% |

Table 6: Energy estimation of ANN (PointNet) and SNNs (Spiking PointNet) of computation.

| Method | Time step | Acc. | #Add. | #Mult. | Energy |
|---|---|---|---|---|---|
| PointNet | - | 92.98% | 0.03M | 13.94M | $1.4 \times 10^8$pJ |
| Spiking PointNet | 1 | 91.66% | 0.45M | 0.45M | $9.2 \times 10^6$pJ |
| | 4 | 93.31% | 1.8M | 1.8M | $3.7 \times 10^7$pJ |

the one without MPP (87.13%) and the vanilla Spiking PointNet (86.70%). Similarly, even with a testing time step of 1, our Spiking PointNet with MPP achieves an accuracy of 87.72%, still superior to the performance of the vanilla one trained with 4 time steps (85.59%).

## 4.2 Energy Efficiency

In this section, we conducted a comprehensive investigation into the hardware efficiency of our proposed framework, with a focus on quantifying energy consumption in computational tasks on ModelNet10. For an ANN model, the dot product operation, or Multiply-Accumulate (MAC) operation, involves both addition and multiplication operations. However, the SNN leverages the multiplication-addition transformation advantage, eliminating the need for multiplication operations in all layers except the first layer. Remarkably, in the absence of spikes, hardware can employ sparse computation to completely avoid addition operations. To estimate energy consumption, we adopted the methodology using 45nm CMOS technology following [26; 39]. The MAC operation in ANN consumes 4.6pJ of energy, while the accumulation operation in SNN requires only 0.9pJ. Notably, in line with our trained-less but learning-more paradigm, we achieved a spike firing rate of 18.7% with $k = 5$. Based on our findings, we computed the energy cost and presented the results in Tab. 6. Our network exhibits remarkable energy efficiency, necessitating only $9.2 \times 10^6$pJ of energy per forward pass, which equates to a 15.2-fold reduction in comparison to conventional ANNs. Moreover, when we conduct inference in four time steps, the performance reaches 93.31%, while the energy required is merely about 3.8 times less than that of its ANN counterpart.

## 5 Conclusion

In this paper, we have presented Spiking PointNet, the first spiking neural network (SNN) specifically designed for efficient deep learning on point clouds. This work was motivated by the tremendous potential of SNNs in energy efficiency and the rising demand for efficient point cloud processing techniques, especially in fields such as autonomous driving and augmented reality. We identified two main challenges hindering the application of SNNs in point cloud tasks: the intrinsic optimization difficulty of SNNs, and the high computational and memory cost of point cloud processing, especially for large time steps. To address these obstacles, we proposed a novel trained-less but learning-more paradigm. This paradigm allows for the training of Spiking PointNet with only a single time step, but is capable of achieving superior performance through multiple time step inference. Theoretical justifications and experimental analysis provided in the paper support our method's effectiveness. Additionally, we introduced a membrane potential perturbation method, which significantly en-

hanced the generalization ability of the Spiking PointNet without increasing computational and storage requirements. Our extensive experiments on multiple datasets, including ModelNet10 and ModelNet40, demonstrated the robustness and superiority of Spiking PointNet. Notably, in certain scenarios, Spiking PointNet was even able to outperform its Artificial Neural Network counterparts, an uncommon achievement in the SNN field.

## Acknowledgment

This work is supported by grants from the National Natural Science Foundation of China under contracts No.12202412 and No.12202413.

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
