# OpenReview forum: "Spiking PointNet: Spiking Neural Networks for Point Clouds"
_NeurIPS.cc/2023/Conference — NeurIPS 2023 poster_

### Official Review · Reviewer_o3KR · 2023-07-01

**Soundness:** 2 fair
**Presentation:** 3 good
**Contribution:** 3 good
**Rating:** 7
**Confidence:** 5

**Summary:**

This paper presents the Spiking PointNet, a novel solution for processing point cloud data. The authors propose a "train-less, learn-more" strategy, which leverages the Surrogate Gradient (SG) method to train the SNN over a single time step, while deploying multiple time steps during the inference phase to optimize performance. The authors provide both theoretical analysis and empirical evidence to demonstrate that their unique strategy surpasses the conventional method of training the SNN over multiple time steps. To further enhance the performance, they introduce a membrane potential perturbation method into their framework.

**Strengths:**

1.The "train-less, learn-more" strategy is interesting.
2.The result is really good on various datasets, even outperforms its ANN counterpart in certain scenarios, an achievement rarely seen in SNNs.
3.The paper is characterized by its clear and concise writing style, complemented by easily comprehensible illustrations.

**Weaknesses:**

1.Section 3.4 seems unclear. If the gradient distributions, as depicted in Figure 4, do not match up as expected, would the membrane potential perturbation method introduced in Section 3.5 still be effective? Please clarify this point.
2.Can this method be applied to point segment task?
3.In table 6, the 10^8 is misspelling. Please correct it.

**Questions:**

Please see weaknesses.

**Limitations:**

I find no potential negative societal impact.

---

> ### Author Rebuttal · Authors · 2023-08-07
>
> We appreciate your time in reviewing our paper and your acknowledgment of our work on point cloud, our novel "train-less, learn-more" approach, and our clear writing complemented by understandable illustrations. Please find our point-by-point responses to your queries as follows.
>
> **Q1**: Section 3.4 seems unclear. If the gradient distributions, as depicted in Figure 4, do not match up as expected, would the membrane potential perturbation method introduced in Section 3.5 still be effective? Please clarify this point.
>
> **A1**: Our Membrane Potential Perturbation (MPP) is independent of gradient distributions. It's designed to counterbalance the loss of residual membrane potential when adopting a trained-less but learning-more strategy with a single timestep. Simply put, MPP ensures the advantages of membrane potential variations persist across multiple timesteps in our "single timestep training" setup. This holds true regardless of gradient specifics, as MPP always functions effectively. We've substantiated this with ModelNet10 dataset experiments for k=0.5, 5, and 20, demonstrating MPP's robustness under different conditions.
>
> | $k$\Timestep | 1 | 2 | 3 | 4 |
> | --- | --- | --- | --- | --- |
> | 0.5 | 87.34% | 88.18% | 89.46% | 90.07% |
> | 5 | 91.66% | 92.98% | 92.98% | 93.31% |
> | 20 | 88.93% | 89.70% | 90.36% | 91.05% |
>
> ---
>
> **Q2**: Can this method be applied to point segment task?
>
> **A2**: Indeed, our method is applicable to point cloud segmentation tasks. As evidenced in the table below, we provide the results of part-segmentation experiments conducted on the ShapeNet dataset. These results include comparisons between PointNet and our Spiking PointNet across various testing time steps.
>
> | Datasets | Methods\Timestep | 1 | 2 | 3 | 4 |
> | --- | --- | --- | --- | --- | --- |
> | ShapeNet | PointNet | 83.63% | - | - | - |
> | ShapeNet | Spiking PointNet | 81.62% | 82.88% | 82.65% | 82.95% |
>
> ---
>
> **Q3**: Misspelling.
>
> **A3**: Errors of symbols and misspellings are possibly caused by typos. We will thoroughly review and correct these in the camera-ready version.
>
> ---
>
> If you have further questions, please do not hesitate to reply to us. Thanks again.

---

> > ### Comment · Reviewer_o3KR · 2023-08-15
> >
> > Thanks for the rebuttal. The response could adress my concern about the effectiveness for the MPP and the generality of the framework for segment task. So I would like to raise my rating to 6. However, I still wonder if the MPP could be transferred to the image classification.

---

> > > ### Author Response · Authors · 2023-08-16
> > > **Response to your second run of concern**
> > >
> > > Thanks for your recognition and kind response first. We have further verified the  generalization of MPP for image classification as your suggestion. We did the experiments on CIFAR-10 using ResNet20 and VGG16 as follows.  It can be seen that the MPP also works well on the image task. We will add this in the next version. Thanks again.
> > >
> > > | Methods\Timestep | 1 | 2 | 3 | 4 |
> > > | --- | --- | --- | --- | --- |
> > > | ResNet20 without MPP | 92.02% | 92.46% | 92.45% | 92.55% |
> > > | ResNet20 with MPP | 91.86% | 92.64% | 93.10% | 93.09% |
> > > | VGG16 without MPP | 93.78% | 94.21% | 94.17% | 94.33% |
> > > | VGG16 with MPP | 94.09% | 94.94% | 94.95% | 94.82% |

---

> > > > ### Comment · Reviewer_o3KR · 2023-08-16
> > > >
> > > > Thanks for the authors for the further responses. All my concerns have been addressed. Considering the first time to apply the SNN to point cloud and the generalization and effectiveness of the method, I support the acceptance of this work.

---

> > > > > ### Author Response · Authors · 2023-08-16
> > > > > **Thanks for the recognition**
> > > > >
> > > > > Thanks for your recognition and kind response. Thanks!

---

### Official Review · Reviewer_zgZ2 · 2023-07-02

**Soundness:** 2 fair
**Presentation:** 3 good
**Contribution:** 2 fair
**Rating:** 4
**Confidence:** 4

**Summary:**

To investigate whether SNN can be suitable for 3D data, this paper proposes a Spiking PointNet for point cloud classification tasks. The authors mainly focus on two key issues: (i) Training SNNs with large time steps; (ii) Reducing the extensive memory and computation cost for the proposed spiking point model. The results on ModelNet10 and ModelNet40 datasets show the effectiveness of the proposed Spiking PointNet.

**Strengths:**

i) The topic of SNN for 3D point cloud classification is novel rather than 2D visual recognition.

ii) The writing is straightforward, clear, and easy to understand.


**Weaknesses:**

i) The two key issues (i.e., training large times steps, and reducing the extensive memory and computation cost) do not seem to be well solved. Table 1 and Table 2 demonstrate that the max time step is 4. Table 6 shows that the energy of Spiking PointNet is only 3.8 times less than of its ANN counterpart. In other words, the Spiking PointNet may bring a huge increase in inference speed than ANN-based architecture, but the reduction in power consumption is almost weak.

ii) The simulation calculation of Spiking PointNet power consumption in Section 4.2 does not seem to be so convincing. The proposed Spiking PointNet are not really deployed on neuromorphic chips (e.g., Loihi2), thus the simple power consumption simulation does not make sense. As far as I know, most existing SNNs cannot be deployed to neuromorphic chips. Please explain whether the proposed Spiking PointNet can be friendly to deploying to neuromorphic chips with lower consumption.


**Questions:**

i) Could you please explain why the proposed SNN method has better performance than the corresponding ANN? I am very curious about this conclusion. I would like more theoretical explanations, especially informational theory, rather than experimental explanations.

ii) Could the proposed Spiking PointNet be deployed on a neuromorphic chip? Please explain from the perspective from the designed structure of Spiking PointNet.

iii) Why did the authors add the MPP module to Spiking PointNet? Perhaps, the MPP is to improve the performance. However, this module is independent and generic for any SNN model. Thus, it is difficult to form a whole with the topic of the proposed Spiking PointNet.


**Limitations:**

No. Some suggestions may be improved for Spiking PointNet: (i) Processing point clouds from an asynchronous way or event-driven SNN; (ii) Leveraging temporal cues from point cloud streams; (iii) Attempting to processing event-based data (i.e., 3D event cloud) from event cameras.

---

> ### Author Rebuttal · Authors · 2023-08-07
>
> Thank you for taking the time to review our paper and for recognizing our contributions in point cloud research, our effective method, and clear writing. We will address your concerns and questions in the following response.
>
> **Q1**: The two key issues (i.e., training large timesteps, and reducing the extensive memory and computation cost) do not seem to be well solved. Table 1 and Table 2 demonstrate that the max time step is 4. Table 6 shows that the energy of Spiking PointNet is only 3.8 times less than that of its ANN counterpart. In other words, the Spiking PointNet may bring a huge increase in inference speed than ANN-based architecture, but the reduction in power consumption is almost weak.
>
> **A1**: Sorry for this confusion. The advantage of SNN is to reduce the power consumption, not to increase the inference speed, even will reduce the speed. This is the difference of SNN and ANN, not our weakness. However, compared to other SNNs, we truly have reduced the time steps and extensive memory. Our contribution is to improve SNNs compared with other SNN works, not to improve the disadvantage of the SNN to compare with the ANN. We will further emphasize this in the next version. Sorry for this confusion again. More details, please see the general response.
>
> ---
>
> **Q2**: The proposed Spiking PointNet are not really deployed on neuromorphic chips (e.g., Loihi2), thus the simple power consumption simulation does not make sense. As far as I know, most existing SNNs cannot be deployed to neuromorphic chips. Please explain whether the proposed Spiking PointNet can be friendly to deploying to neuromorphic chips with lower consumption.
>
> **A2**: Thanks for this suggestion. The spiking PointNet is only composed of convolution layers, linear layers, BN layers (which can be folded to the convolution layers), LIF layers, and average pooling layers, which are the commonly used modules that have be proven could be deployed to the Loihi in [1][2], and so on. The only obstacle is that limited by the development of the neuromorphic hardware, it is not easy to deploy a large model on them. However, this is also the advantage of the whole SNN domain. On top of that, we provide a relatively fair comparison as in other SNN works.
>
> [1] LaneSNNs: Spiking Neural Networks for Lane Detection on the Loihi Neuromorphic Processor, IROS 2022.
>
> [2] An Efficient Spiking Neural Network for Recognizing Gestures with a DVS Camera on the Loihi Neuromorphic Processor, IJCAI 2020.
>
> ---
>
> **Q3**: Could you please explain why the proposed SNN method has better performance than the corresponding ANN? I am very curious about this conclusion. I would like more theoretical explanations, especially informational theory, rather than experimental explanations.
>
> **A3**: Thanks for the question. The ANN can be divided into two parts as in many works like [1,2], the first part can be viewed as a feature encoder, $f_e$, and the second part can be viewed as a classifier, $f_c$. The output feature is a $C \times 1$ vector, computed by $f_e(x)$. It is shown crucial that output feature representation is rich and powerful for training a highly accurate model for the classification in many recent works like [1]. For the PointNet, the representation output feature of SNN has a chance to outperform the ANN. Take the information entropy concept to analyze it, given a set, $M$, its representation capability, $\mathcal{R}(M)$ can be measured by the information entropy of $M$, as follows
> $$
> \mathcal{R}(M) = \max \mathcal{H}(M) = \max (-\sum_{m \in M } p_M(m){ log}p_M(m)),
> $$
> where $p_M(m)$ is the probability of a sample, $m$ from $ M$. When $p_M(m_1)=p_M(m_2)=\cdots = p_M(m_N)$, $\mathcal{H}(M)$ reaches its maximum, $log(N), $where $N$ is the total number of the samples from $M$. Since a single channel needs 32 bits to represent, consisting of $2^{32}$ samples, the $\mathcal{R}(M) = log(2^{32\times C}) = 32\times C$ for the output feature of ANN, where C is the channel number. While for one timestep of the SNN, the output feature comes from the average pooling of 1024 points. One point output a binary spike, thus the average pooling needs $log(1024) = 10$ bits to represent.  Considering the multiple timesteps, it will increase to $10 \times T$ bits. the $\mathcal{R}(M) = log(2^{10 \times T \times C}) = 10 \times T \times C$ for SNN, where T is timesteps. With the increasing of T, the representation output feature of SNN has a chance to outperform the ANN.
>
> But for the image task, like CIFAR, the average pooling only needs $log(4) = 2$ bits to represent, with the increasing of T, its training difficulty will increase greatly too. Hence it is not easy to outperform ANN in image tasks.
>
> [1] A Simple Framework for Contrastive Learning of Visual Representations.
>
> [2] Learning Transferable Visual Models From Natural Language Supervision.
>
> ---
>
> **Q4**: Could the proposed Spiking PointNet be deployed on a neuromorphic chip?
>
> **A4**: Thanks for the question. Please see our response to **Q2**.
>
> ---
>
> **Q5**: Why did the authors add the MPP module to Spiking PointNet?  it is difficult to form a whole with the topic of the proposed Spiking PointNet.
>
> **A5**: Thanks for the question. The residual membrane potential along time steps in the spiking neuron for point cloud can be seen as the perturbation which can improve the uncertainty estimation and thus may lead to an enhancement in SNN accuracy, see line 224-228 in our paper. However, to solve the expensive memory of deep learning for point cloud, we propose the trained-less but learning-more paradigm, which will lose the residual membrane potential perturbation with only a single time step. Then to retain the residual membrane potential perturbation advantage for the multiple timesteps again, we propose and embed the MMP method in the proposed trained with one timestep paradigm.
>
> ---
>
> If you have further questions, please do not hesitate to reply to us. Thanks again.

---

> > ### Comment · Reviewer_zgZ2 · 2023-08-19
> > **Thanks for the rebuttal**
> >
> > Thank you for the authors' response. I still have reservations regarding the deployment of the proposed method on neuromorphic chips. Additionally, I remain skeptical about the conclusion that the SNN method performs better than the corresponding ANN. If this conclusion is valid, it would be groundbreaking news in the field of artificial intelligence, particularly in neuromorphic computing. Taking all these factors into consideration, I will uphold my original score of Borderline Reject for this work.

---

> > > ### Author Response · Authors · 2023-08-19
> > > **Thank for your second run of comments**
> > >
> > > Very thanks for your response. We are happy to find that most questions have been addressed by comparing your first and second runs of the comments, but still two concerns. From these two concerns, we can see your professionalism since these concerns you proposed are two fundamental issues that confuse most SNN researchers still, not only exist in our works. We really hope to have further opportunities to discuss these concerns with you in depth.
> > >
> > > **About if there is a chance that SNN can outperform the ANN sometimes.** As you said, most prior work shows it is impossible and it seems to be a consensus in the SNN filed. I can understand your disappointment with this consensus since we are too. However, in this work, we find it is possible and we have provided the detailed code for both ANN and SNN and instructions to verify. We also provide a theoretical explanation in the first run of the response. **We notice there is also some work [1] reports the same phenomenon (See the result on CIFAR with ResNet19). In fact, in almost all these recent quantization works, with 2- or 3-bit weights even, this has been very common [2,3].** With higher precision weights and more timesteps than QNNs, we believe SNNs could also surpass ANNs with a certain probability. Please give the SNN field some confidence. We believe you also hope to see that the SNN can replace or keep pace with ANN one day. However, this needs you, us, and other SNN researchers to keep enthusiasm and try hard. **So, please don't easily negate our results, and we sincerely hope you could take some time to check our code to verify the phenomenon.**
> > >
> > > **About most existing SNNs cannot be deployed to neuromorphic chips.** We partially agree with you. This is a huge obstacle for the SNN field. However, we still hope you could keep some confidence and patience for the field. Like QNN, with the development, more attention will be paid to this field, then the practical application will be accelerated. In fact, there have been many recent SNN works that have closed the accuracy gap between SNN and ANN much. And there are some hardware work that can run SNN like ANN, e.g., Tianjin [4]. **Thus, we sincerely hope you could temporarily put aside these temporary limitations of SNN, and reconsider our contributions to the field compared to other SNN works again.**
> > >
> > > Sincerely hope for your further response. Thank you very much for your time and efforts again!
> > >
> > > [1] Guo Y , Chen Y , Zhang L ,et al.Reducing Information Loss for Spiking Neural Networks[C]//European Conference on Computer, 2022
> > >
> > > [2] Liu, Zechun , et al. "Nonuniform-to-Uniform Quantization: Towards Accurate Quantization via Generalized Straight-Through Estimation." arXiv e-prints, 2022
> > >
> > > [3] Esser S , Mckinstry J L , Bablani D ,et al. LSQ:Learned Step Size Quantization, ICLR 2020
> > >
> > > [4] Pei, J., Deng, L., Song, et al.: Towards artificial general intelligence with hybrid Tianjin chip
> > >
> > > architecture. Nature 572(7767), 106–111 (2019）

---

> ### Author Response · Authors · 2023-08-17
> **Waiting for feedback**
>
> Dear Reviewer zgZ2,
>
> Since the author-reviewer discussion period is approaching the deadline, we would appreciate it if you could check our response to your review comments soon. This way, if you have further questions and comments, we can still reply before the author-reviewer discussion period ends. Thank you very much for your time and efforts!
>
> Best,
>
> The authors

---

### Official Review · Reviewer_CQRm · 2023-07-05

**Soundness:** 2 fair
**Presentation:** 2 fair
**Contribution:** 2 fair
**Rating:** 4
**Confidence:** 5

**Summary:**

The authors replaced the ReLU activation function of the original PointNet with the LIF neurons and named it spiking PointNet. The work offers trained-less but learning-more tricks and corresponding analysis. The effectiveness of spiking PointNet is evaluated on the ModelNet10/40 datasets, specifically in point cloud classification.

**Strengths:**

In general, the submission is well-written and easy to follow.

**Weaknesses:**

-1. If I understand correctly, the authors only replaced ReLu with LIF. If so, whether the first or not does not make sense as we can make first spiking VGG-16, first spiking ResNet, etc.

-2. It is unclear to me what's the output of a LIF neuron. Is it a potential value or spiking? I guess the authors used potential value.

-3. How does the leaky part of LIF impact the spiking PointNet? The authors claimed inference multiple times offers better performance. I am wondering, if we input a point cloud multiple times with different time intervals, what are the performance differences?

-4. What's the processing speed with different T?

-5. The validations are only with ModelNet. I think validations with more datasets are necessary, such as ShapeNet dataset.

**Questions:**

Please see weakness section

**Limitations:**

No limitation is discussed.

If the work only replaces ReLU with LIF, I think the authors should show whether the replacement and training tricks would work with other models, such as VGG, and ResNet.

---

> ### Author Rebuttal · Authors · 2023-08-07
>
> Thank you for your efforts in reviewing our paper and your recognition of our effective method and well-written submission. The response to your questions is given piece by piece as follows.
>
> ---
>
> **Q1**: If I understand correctly, the authors only replaced ReLu with LIF. If so, whether the first or not does not make sense as we can make first spiking VGG-16, first spiking ResNet, etc.
>
> **A1**: Sorry for this confusion. Since it still remains underexplored whether SNNs can be generalized to 3D recognition, we try to apply the SNN to deep learning on point clouds. We then found it challenging to train a big spiking model with large time steps for point cloud due to that the intrinsic optimization of the SNN with large time steps, and the expensive memory and computation cost of deep learning for point cloud that makes training a big spiking point model unrealistic. To solve the problems simultaneously, we propose a trained-less but learning-more paradigm and a membrane potential perturbation method. We will emphasize our attempt, findings, train of thought, and the solution more in the final version to highlight our contributions while weaken the statement that the first spiking neural network approach to deep learning on point clouds. Thanks for your suggestion.
>
> In addition, we guess you may think the trained-less but learning-more paradigm is a general framework that could be applied to other networks and domains, thus it can not be seen the first SNN for point cloud. I partly agree with this. However, the trained-less but learning-more paradigm is a counter-intuitive finding. The more timesteps, the higher accuracy seems have been a consensus in the prior works. Hence, someone may have already tried to apply the SNN to point clouds in the past. However, because deep learning for point clouds is memory-intensive and the challenges posed by large timesteps, they were unable to train the SNN effectively, especially when constrained by limited hardware. We also face the same problem, but with theoretical justifications and in-depth experimental analysis, we find that training the SNN with only a single timestep is better and can solve the problem of applying SNN to point cloud. On top of that, our method can also be used for other domains and networks, this shows the novelty of our train of thought to try for point cloud first and then the generalization of our method.
>
> ---
>
> **Q2**: The output of a LIF neuron.
>
> **A2**: We apologize for the confusion. The output of a LIF layer is a binary spike map. It could be found in Eq.3 in our paper and line 40-70 in spike_layer_without_MPR.py in our provided code. We will clearly highlight the output in the final version as your suggestion. Thanks.
>
> ---
>
> **Q3**: How does the leaky part of LIF impact the spiking PointNet? If we input a point cloud multiple times with different time intervals, what are the performance differences?
>
> **A3**: Thanks for the question. For the leaky part, we add the ablation study with different leaky values on ModelNet10 as below. It can be seen that the leaky should not be too small or large. When it $\in [0.15,0.3]$, the performance is relatively better. We will add the findings as your suggestion in the next version. Thanks.
>
> | Leaky part \ Timestep | 1 | 2 | 3 | 4 |
> | --- | --- | --- | --- | --- |
> | 0.05 | 90.59% | 90.72% | 91.21% | 91.36% |
> | 0.1 | 90.61% | 90.99% | 90.94% | 91.59% |
> | 0.15 | 91.05% | 92.23% | 92.73% | 92.96% |
> | 0.2 | 91.74% | 92.53% | 93.07% | 93.26% |
> | 0.25 | 91.66% | 92.98% | 92.98% | 93.31% |
> | 0.3 | 91.22% | 92.75% | 93.21% | 93.28% |
> | 0.35 | 90.82% | 91.15% | 91.75% | 92.38% |
> | 0.5 | 89.67% | 90.02% | 91.39% | 91.43% |
>
> As for the different time intervals, we add the ablation study on ModelNet10 further show the results of every time step and different time intervals as below. It can be seen with multiple times with different time intervals, it is all better than the single one time step, and with more timesteps is some slightly better than the less one in more times. We will add the findings as your suggestion in the next version. Thanks.
>
> |  Timestep | 1th | 2th | 3th | 4th | 5th | 6th | 7th | 8th |
> | --- | --- | --- | --- | --- | --- | --- | --- | --- |
> | Accuracy | 91.66% | 91.21% | 91.65% | 90.85% | 91.43% | 91.77% | 91.18% | 90.35% |
> | Time interval | 1-2 | 2-3 | 3-4 | 4-5 | 5-6 | 6-7 | 7-8 |  |
> | Accuracy | 92.98% | 92.17% | 91.96% | 92.01% | 92.96% | 93.08% | 91.75% |  |
> | Time interval | 1-4 | 2-5 | 3-6 | 4-7 | 5-8 |  |  |  |
> | Accuracy | 93.31% | 93.24% | 93.42% | 93.27% | 93.19% |  |  |  |
>
> ---
>
> **Q4**: What's the processing speed with different T?
>
> **A4**: Thanks for the question. We add the time for 1 epoch inference with batchsize as 48 on ModelNet10 based on a single 3090 as follows. More details, please see our general response. We will add this in the next version. Thanks.
>
> | Timestep | 1 | 2 | 3 | 4 |
> | --- | --- | --- | --- | --- |
> | Time(s) | 0.3934 | 0.4758 | 0.5930 | 0.6624 |
>
> ---
>
> **Q5**: Validations with ShapeNet dataset.
>
> **A5**: Thanks for the question. We add the experiments on ShapeNet as follows. It also clearly shows our method’s effectiveness. We will add this in the next version. Thanks.
>
> | Methods\Timestep | 1 | 2 | 3 | 4 |
> | --- | --- | --- | --- | --- |
> | PointNet | 83.63% | - | - | - |
> | Spiking PointNet | 81.52% | 82.88% | 82.65% | 82.95% |
>
> ---
>
> **Q6**: Whether the replacement and training tricks would work with other models, such as VGG, and ResNet.
>
> **A6**: Thanks for your suggestion, we also add the experiments on CIFAR-10 using ResNet20 and VGG16 as follows.  It shows our method’s generalization. We will add this in the next version. Thanks.
>
> | Methods\Timestep | 1 | 2 | 3 | 4 |
> | --- | --- | --- | --- | --- |
> | ResNet20  | 91.86% | 92.64% | 93.10% | 93.09% |
> | VGG16 | 94.09% | 94.94% | 94.95% | 94.82% |
>
> ---
>
> If you have further questions, please do not hesitate to reply to us. Thanks again.

---

> ### Author Response · Authors · 2023-08-17
> **Waiting for feedback**
>
> Dear Reviewer CQRm,
>
> Since the author-reviewer discussion period is approaching the deadline, we would appreciate it if you could check our response to your review comments soon. This way, if you have further questions and comments, we can still reply before the author-reviewer discussion period ends. Thank you very much for your time and efforts!
>
> Best,
>
> The authors

---

### Official Review · Reviewer_f5wg · 2023-07-06

**Soundness:** 3 good
**Presentation:** 2 fair
**Contribution:** 3 good
**Rating:** 6
**Confidence:** 4

**Summary:**

In this paper, the authors introduce the spiking neural network into 3D data processing, the point cloud data. In specific, the authors analyze the training difficulty when using SNNs for the point cloud recognition and find that number of timesteps should not be increased in the training phase. Their framework, the trained-less but learning more, obtains better performance than aligning the number of timesteps in training and inference.

**Strengths:**

+ Well-studied SNNs on the point cloud dataset for the first time. This paper brings the advantage of energy efficiency of neuromorphic computing to 3D vision topic.

+ Interesting, counter-intuitive findings but a surprisingly well-performance method.


**Weaknesses:**

- The authors claimed theoretical analysis in Section 3.4. However, I found it mostly empirical analysis. I wonder where is the theory?

- What will happen if the timesteps are increased more than 4 during inference?


**Questions:**

Is it possible to convert an ANN PointNet to SNN?

**Limitations:**

See above for improvements

---

> ### Author Rebuttal · Authors · 2023-08-07
>
> Thanks for your efforts in reviewing our paper and your recognition of our interesting findings, efforts for point cloud,  and the well-performance method. The response to your questions is given piece by piece as follows.
>
> ---
>
> **Q1**: The authors claimed theoretical analysis in Section 3.4. However, I found it mostly empirical analysis. I wonder where is the theory.
>
> **A1**: Thanks for this advice. We present the proof to show that larger time steps will make the training of SNNs more difficult as follows.
>
> Proof.
>
> The gradients to weights have been given in our paper, as follows:
> $$
> \frac{\partial L}{\partial \mathbf{W}^l}=\sum_{t=1}^T \frac{\partial L}{\partial \mathbf{s}^{l+1}[t]} \textcolor{red}{\frac{\partial \mathbf{s}^{l+1}[t]}{\partial \mathbf{u}^{l+1}[t]}}\left(\frac{\partial \mathbf{u}^{l+1}[t]}{\partial \mathbf{W}^l}+\sum_{\tau<t} \prod_{i=t-1}^\tau\left(\textcolor{green}{\frac{\partial \mathbf{u}^{l+1}[i+1]}{\partial \mathbf{u}^{l+1}[i]}}+\textcolor{blue}{\frac{\partial \mathbf{u}^{l+1}[i+1]}{\partial \mathbf{s}^{l+1}[i]}} \textcolor{red}{\frac{\partial \mathbf{s}^{l+1}[i]}{\partial \mathbf{u}^{l+1}[i]}}\right) \frac{\partial \mathbf{u}^{l+1}[\tau]}{\partial \mathbf{W}^l}\right)
> $$
> Since the loss can be computed by:
>
> $$
> L = \frac{1}{T}\sum_{t=1}^{T} L_{CE}(y,o[t]).
> $$
> the gradients to weights for large T and small T can be further updated as:
> $$
> \frac{\partial L^{large}}{\partial \mathbf{W}^l}=\sum_{t=1}^{T^{large}} \frac{\partial L_{CE} / T}{\partial \mathbf{s}^{l+1}[t]} \textcolor{red}{\frac{\partial \mathbf{s}^{l+1}[t]}{\partial \mathbf{u}^{l+1}[t]}}\frac{\partial \mathbf{u}^{l+1}[t]}{\partial \mathbf{W}^l}+\sum_{t=1}^T \frac{\partial L_{CE} / T}{\partial \mathbf{s}^{l+1}[t]} \textcolor{red}{\frac{\partial \mathbf{s}^{l+1}[t]}{\partial \mathbf{u}^{l+1}[t]}}\left(\sum_{\tau<t} \prod_{i=t-1}^\tau\left(\textcolor{green}{\frac{\partial \mathbf{u}^{l+1}[i+1]}{\partial \mathbf{u}^{l+1}[i]}}+\textcolor{blue}{\frac{\partial \mathbf{u}^{l+1}[i+1]}{\partial \mathbf{s}^{l+1}[i]}} \textcolor{red}{\frac{\partial \mathbf{s}^{l+1}[i]}{\partial \mathbf{u}^{l+1}[i]}}\right) \frac{\partial \mathbf{u}^{l+1}[\tau]}{\partial \mathbf{W}^l}\right)
> $$
>
> and
>
> $$
> \frac{\partial L^{small}}{\partial \mathbf{W}^l}=\sum_{t=1}^{T^{small}} \frac{\partial L_{CE} / T}{\partial \mathbf{s}^{l+1}[t]} \textcolor{red}{\frac{\partial \mathbf{s}^{l+1}[t]}{\partial \mathbf{u}^{l+1}[t]}}\frac{\partial \mathbf{u}^{l+1}[t]}{\partial \mathbf{W}^l}+\sum_{t=1}^T \frac{\partial L_{CE} / T}{\partial \mathbf{s}^{l+1}[t]} \textcolor{red}{\frac{\partial \mathbf{s}^{l+1}[t]}{\partial \mathbf{u}^{l+1}[t]}}\left(\sum_{\tau<t} \prod_{i=t-1}^\tau\left(\textcolor{green}{\frac{\partial \mathbf{u}^{l+1}[i+1]}{\partial \mathbf{u}^{l+1}[i]}}+\textcolor{blue}{\frac{\partial \mathbf{u}^{l+1}[i+1]}{\partial \mathbf{s}^{l+1}[i]}} \textcolor{red}{\frac{\partial \mathbf{s}^{l+1}[i]}{\partial \mathbf{u}^{l+1}[i]}}\right) \frac{\partial \mathbf{u}^{l+1}[\tau]}{\partial \mathbf{W}^l}\right)
> $$
>
> To simplify the proof, $\frac{\partial L_{CE} / T}{\partial \mathbf{s}^{l+1}[t]}$, $\frac{\partial \mathbf{s}^{l+1}[t]}{\partial \mathbf{u}^{l+1}[t]}$, etc, can be viewed as identical. Obviously, the first stem for the above two equations are same, while the second stem for the large T is larger.  It can also be easily understood when combined with Figure 2 in our paper. That is to say $\sum_{\tau<t} \prod_{i=t-1}^\tau\left(\textcolor{green}{\frac{\partial \mathbf{u}^{l+1}[i+1]}{\partial \mathbf{u}^{l+1}[i]}}+\textcolor{blue}{\frac{\partial \mathbf{u}^{l+1}[i+1]}{\partial \mathbf{s}^{l+1}[i]}} \textcolor{red}{\frac{\partial \mathbf{s}^{l+1}[i]}{\partial \mathbf{u}^{l+1}[i]}}\right) \frac{\partial \mathbf{u}^{l+1}[\tau]}{\partial \mathbf{W}^l}$, denoted as **G**, will contribute more to the large T.
>
> As we have mentioned in our paper, the $\textcolor{red}{\frac{\partial \mathbf{s}^{l+1}[i]}{\partial \mathbf{u}^{l+1}[i]}}$ can be controlled by a $k$. When $k$ is large, a more accurate gradient in the backward pass can be obtained, while the gradient will be sharp at a narrow range while very small in the residual part. Thus the gradient explode or vanish problem will become severe in this case. Due to the multiple connected multiplication characteristic of **G**, the gradient explode or vanish problem will be more severe. While $k$ is small,  an inaccurate gradient in the backward pass will be obtained. Hence more gradient error will be accumulated through time steps based on the multiple connected accumulation characteristic of **G**, thus hurting the performance of the SNN too.
>
> Consequently, it is very difficult to train a well-performed SNN with large timesteps directly.
>
> ---
>
> **Q2**: What will happen if the timesteps are increased more than 4 during inference?
>
> **A2**: Thanks for the question. With the timesteps increasing, the accuracy will converge to a relatively high value, better than the single time step. We provide more results on the ModelNet10 and ModelNet40 as below.
>
> | Datasets\Timestep | 1 | 2 | 3 | 4 | 5 | 6 | 7 | 8 |
> | --- | --- | --- | --- | --- | --- | --- | --- | --- |
> | ModelNet10 | 91.66% | 92.98% | 92.98% | 93.31% | 92.94% | 93.37% | 93.28% | 93.21% |
> | ModelNet40 | 87.72% | 88.46% | 88.25% | 88.61% | 88.41% | 88.73% | 88.20% | 88.59% |
>
> **Q3**: Is it possible to convert an ANN PointNet to SNN?
>
> **A3**: Yes. The PointNet is also composed of convolution layers, linear layers, BN layers, and Pooling layers as ResNet, which have been proven could be converted to SNN in these ANN-SNN works.
>
> ---
>
> If you have further questions, please do not hesitate to reply to us. Thanks again.

---

> > ### Comment · Reviewer_f5wg · 2023-08-12
> > **Thanks for the rebuttal**
> >
> > I would like to thank the authors for their reply. It is interesting to see the accuracy maintains a high value even after increasing the number of timesteps to 8. Based on this and the value of being the first to investigate the SNNs in the 3D vision area, I would recommend a publication at NeurIPS. Therefore, I increase my score to 6.

---

> > > ### Author Response · Authors · 2023-08-16
> > > **Thanks for the recognition**
> > >
> > > Thanks for your recognition and kind response. Thanks!

---

### Author Rebuttal · Authors · 2023-08-07

We thank all reviewers’ time and effort in reviewing our paper.  While we will respond to the comments piece by piece, we would like to first provide a general response to the common concern about the inference speed of spiking PointNet.

The advantage of SNNs is to reduce the power consumption, not to increase the inference speed and even will reduce the speed. This is the difference of SNN and ANN, not our weakness. However, compared to other SNNs, we have significantly reduced the time steps and extensive memory usage. Our contribution is to improve SNNs compared with other SNN works, not to improve the disadvantage of the SNN to compare with the ANN. We will further emphasize this in the next version. Sorry for this confusion again.

In addition, the integration of storage and computation paradigm and the asynchronous parallel processing on neuromorphic hardware, like Darwin[1], Tianjic[2], and other memristor-enabled neuromorphic computing systems[3], makes SNNs much different from ANNs. In these hardware, once the previous time step map has been processed by one layer and presented to the next layer to deal with, the layer would process the next time step data right after that while the next layer processes the presented map. Due to this kind of asynchronous parallel processing paradigm, the inference speed is almost unaffected with the time step increasing.

Even on the GPU, the reduction in speed is not linear with the increase in timesteps. We add the time for 1 epoch inference with batch size as 48 on ModelNet10 based on a single 3090 as follows. The inference time for 4 timesteps is only 1.68 times to 1 timesteps.

| Timestep | 1 | 2 | 3 | 4 |
| --- | --- | --- | --- | --- |
| Time(s) | 0.3934 | 0.4758 | 0.5930 | 0.6624 |

[1] Darwin: A neuromorphic hardware co-processor based on spiking neural networks.

[2] Towards artificial general intelligence with hybrid tianjic chip architecture. Nature.

[3] Fully hardware-implemented memristor convolutional neural network. Nature.

---

### Decision · Program_Chairs · 2023-09-21

**Decision:**

Accept (poster)

**Comment:**

The reviewers mentioned several positive aspects of the paper, e.g.:
- Interesting, counter-intuitive findings but a surprisingly well-performance method.
- The paper is well-written and easy to follow.
- The "train-less, learn-more" strategy is interesting.
- The result is really good on various datasets, even outperforms its ANN counterpart in certain scenarios, an achievement rarely seen in SNNs.

However, they also raised multiple doubts about it, such as:
- The presented theoretical analysis in the rebuttal should be included in the appendix.
- Some parts of the paper (e.g., replacing LIF with ReLUs, outputs of a LIF neuron) are unclear.
- The two key issues (i.e., training large time steps, and reducing the extensive memory and computation cost) do not seem to be well solved.

The paper is borderline (7-6-4-4), however, the reviewers did a great job in the rebuttal to address all issues raised by the reviewers. Some reviewers still have som reservations, but overall the paper is well-written and some confusing parts could be easily addressed in the appendix. As a result, I suggest accepting the paper.